# Review of Gold Nanoparticles: Synthesis, Properties, Shapes, Cellular Uptake, Targeting, Release Mechanisms and Applications in Drug Delivery and Therapy

**DOI:** 10.3390/pharmaceutics16101332

**Published:** 2024-10-16

**Authors:** Joel Georgeous, Nour AlSawaftah, Waad H. Abuwatfa, Ghaleb A. Husseini

**Affiliations:** 1Biomedical Engineering Program, College of Engineering, American University of Sharjah, Sharjah P.O. Box 26666, United Arab Emirates; g00099462@aus.edu; 2Materials Science and Engineering Ph.D. Program, College of Arts and Sciences, American University of Sharjah, Sharjah P.O. Box 26666, United Arab Emirates; g00051790@alumni.aus.edu (N.A.); g00062257@alumni.aus.edu (W.H.A.); 3Department of Chemical and Biological Engineering, College of Engineering, American University of Sharjah, Sharjah P.O. Box 26666, United Arab Emirates; 4Biosciences and Bioengineering Ph.D. Program, College of Engineering, American University of Sharjah, Sharjah P.O. Box 26666, United Arab Emirates

**Keywords:** gold nanoparticles, drug delivery, nanotechnology, cancer treatment, targeted delivery

## Abstract

The remarkable versatility of gold nanoparticles (AuNPs) makes them innovative agents across various fields, including drug delivery, biosensing, catalysis, bioimaging, and vaccine development. This paper provides a detailed review of the important role of AuNPs in drug delivery and therapeutics. We begin by exploring traditional drug delivery systems (DDS), highlighting the role of nanoparticles in revolutionizing drug delivery techniques. We then describe the unique and intriguing properties of AuNPs that make them exceptional for drug delivery. Their shapes, functionalization, drug-loading bonds, targeting mechanisms, release mechanisms, therapeutic effects, and cellular uptake methods are discussed, along with relevant examples from the literature. Lastly, we present the drug delivery applications of AuNPs across various medical domains, including cancer, cardiovascular diseases, ocular diseases, and diabetes, with a focus on in vitro and in vivo cancer research.

## 1. Introduction

The administration of a pharmaceutical compound to achieve a therapeutic response in an organism, referred to as drug delivery [1], encompasses several methods such as oral [2], rectal [3], parenteral (subcutaneous, intramuscular, intravenous, or intra-arterial injections) [4], transmucosal [5], pulmonary [6], transdermal [7], and intraocular delivery systems [8]. Oral administration has historically been the most commonly used due to its ease of administration, controlled delivery, and patient acceptance [2]. Although rectal delivery can cause discomfort to patients and has a small absorption area, it is suitable for local interventions due to low rectal enzymatic activity [3]. While injections are invasive and can be painful, making them less preferred by patients, they are effective when oral medication requires a high dosing frequency or is ineffective due to the acidic nature of the gastrointestinal tract. Parenteral drug delivery methods ensure maximum drug absorption and provide a shorter treatment period [4]. Transmucosal drug delivery is painless, flexible, and convenient for rapid drug uptake, as transmucosal membranes are relatively permeable with rich blood flow. However, this route is limited to low-dose drugs that undergo passive diffusion and remain stable at buccal pH [5]. Pulmonary drug delivery systems are based on inhalation therapy and utilize lung properties such as high absorptive surface area and intense blood supply. Some challenges to this technique include low drug dosages, poor drug stability, and the low efficiency of the inhalation practice [6]. Transdermal drug delivery is a painless method that involves applying a drug formulation to dry skin, allowing it to penetrate the epidermis and reach the dermal microcirculation. However, the stratum corneum, the outermost skin layer, poses the most significant challenge as it can prevent the permeation of foreign compounds [7]. To reduce repeated drug injections for treating retinal diseases, an intraocular drug delivery system offers a noninvasive technique that can release the drug over a long period with a single administration. However, the retinal blood barrier can block foreign compounds from entering the eye, limiting the therapeutic effect of this method [8]. As shown in Table 1, the effectiveness of traditional DDSs depends on the route of administration, as barriers such as pH, mucus layers, enzymes, and other physiological conditions negatively impact the administered drugs. Therefore, with traditional DDSs, drugs are absorbed less efficiently, may cause side effects, and often take longer to cure the targeted disease.

### Nanoparticles

The inefficiency of traditional DDSs led to the novel development of controlled drug delivery through nanoparticles, enhancing the process in several ways [9]. Nanotechnology is the study of structures that range in size at the nanoscale level (1–100 nm) [10]. Nanoparticles (NPs) exhibit unique and interesting properties due to their very small size, making them highly useful in various fields, such as materials engineering, environmental science, electronics, energy harvesting, mechanical industries, and biomedical engineering [11]. NPs offer a wide range of applications in the health field, including biosensing, bioimaging, gene therapy, antimicrobial materials, tissue engineering, and drug delivery [12]. Due to their nanoscale size, NPs are extensively used in drug delivery, as they provide the ability to deliver drugs to a specific area, organ, or tissue in the body, enhancing stability, prolonging the drug’s effect, reducing immune recognition, and minimizing side effects [13]. Composed of biocompatible materials, NPs ensure minimal side reactions and toxicity in the body. Table 1 illustrates the stark contrast between traditional and nanoparticle-based DDSs.

Due to their unique properties, NPs offer several advantages over traditional DDSs. First, their small size enables them to interact at the cellular level with biological structures and to pass through tissues and other barriers in the body. Additionally, their large surface area allows for a greater drug-loading capacity. Since their surface can be modified with various functional groups, NPs can precisely target specific biological entities. This makes them highly specific, targeting particular cells, tissues, and organs, thus minimizing off-target effects. More importantly, NPs enable controlled drug release at the target location, ensuring a consistent therapeutic effect and reducing the need for frequent dosing. Moreover, NPs can protect drugs from degradation, enhancing their stability [14].

**Table 1 pharmaceutics-16-01332-t001:** Different Properties of traditional DDSs and nanoparticle-based DDSs.

Properties	Traditional DDSs	Nanoparticle-Based DDSs
Cellular-level Interaction	Cannot always cross specific tissues and barriers (i.e., the blood–brain barrier, the skin, GI tract) [2,3,7]	Surface structure can be modified to pass tissues and other barriers [14]
Target and Specificity	Administered to the whole body	Can be administered to a specific biological entity [14]
Side Effects	The lack of specificity might cause various side effects in the body [14]	High specificity will minimize interactions with other body organs tissues, or cells, minimizing side effects [14]
Dosage	The route can limit drug dosage [5,6]	Delivers drugs in the optimum dosage [11]
Consistency	Does not provide a constant therapeutic effect [2]	Provides a constant therapeutic effect [14]
Stability	The drug is not protected and can suffer degradation due to various body internal environmental factors [2]	NPs protect drugs from degradation, increasing their shelf life [14]
Patient Compliance	Lower patient compliance [3]	Improved patient compliance [11]

This paper provides a comprehensive review of AuNPs, which are increasingly popular for their diverse applications in drug delivery, particularly as efficient and safe carriers for anticancer drugs. Their appeal lies in their tunable size and shape, unique optical properties, chemical stability, and the ability to modify their surface functionalities, enabling their use across a broad spectrum of diseases and applications. AuNPs also possess advantageous properties, such as being inert, non-toxic, and biocompatible, making them ideal carriers for therapeutic drugs. Their synthesis can be customized to produce particle sizes comparable to biomolecules, enhancing their integration within living organisms. More importantly, their high surface area-to-volume ratio allows for tunable functionalization, improving their ability to bind to targets and therapeutic drugs [15]. Figure 1 summarizes the key milestones in colloidal chemistry, particularly in the development of AuNPs based on the historical timeline described by [16]. 

Despite the various applications of AuNPs in genomics, biosensors, clinical chemistry, and cellular-level optical bioimaging [15], this paper will focus on the comprehensive attributes of AuNPs in drug delivery. It will also thoroughly discuss the different synthesis methods, physicochemical properties, shapes, loading methods, targeting techniques, and release mechanisms of AuNPs, as well as their application in therapeutics and drug delivery, along with related case studies. Various attributes, such as size, shape, and surface chemistry, can significantly affect the wide-range applications of AuNPs, influencing their behavior in biological settings, which this review will examine.

## 2. Properties

AuNPs possess various properties that make them suitable for medical applications, including drug delivery, photomedicine, tissue engineering, biosensing, antimicrobial agents, anticancer agents, and cellular probes [17]. Nevertheless, this paper will focus on the properties of AuNPs that make them suitable candidates for drug delivery and therapeutic applications. These properties include size, shape, and surface characteristics, which influence their behavior, such as cellular uptake, specificity, toxicity, and ability to permeate barriers in the body. Additionally, features like a large surface area-to-volume ratio, high plasmonic resonance, multi-functionalization, reproducibility, and stability make AuNPs promising drug carriers [18].

One of the most advantageous properties of AuNPs is their ease of surface functionalization, which allows them to be customized for therapeutic applications. As will be discussed in more detail in a later section, AuNPs can be modified with various molecules, including drugs, antibodies, peptides, and polymers [19]. Functionalization can provide colloidal stability, preventing the NPs from aggregating and clumping, thereby ensuring they function as intended [20]. In addition, surface functionalization helps reduce toxicity and enhances the biocompatibility of AuNPs. It also enables AuNPs to carry particles, such as drugs, to the targeted diseased site [21]. The particles can precisely reach their target location because they can be functionalized with moieties such as antibodies, peptides, and hormones that specifically target desired cells, such as cancer cells [19]. In addition, the size and shape of AuNPs determine their ability to accumulate in tumors through the enhanced permeability and retention (EPR) effect [22]. Additionally, the small size of AuNPs enables their internalization into targeted cells through endocytosis. A size range of 40–60 nm has been shown to provide the highest efficiency in cellular uptake [23]. AuNPs with a size smaller than 3 nm can translocate through the cell’s lipid bilayer; however, particles of this size are often easily eliminated through renal clearance [24]. AuNPs can also serve as water-soluble vehicles for delivering hydrophobic drugs [25]. The impact of each property on the behavior of AuNPs during the drug delivery process will be discussed and explained further in this paper.

Another notable property of AuNPs that is of paramount importance is localized surface plasmon resonance (LSPR), which results from the oscillation of free electrons at the nanoscale. When light strikes these particles, it interacts with their molecules, gaining energy and causing the light to rotate and vibrate differently. This phenomenon is known as surface-enhanced Raman scattering (SERS), primarily driven by electromagnetic enhancement [26]. This enhancement occurs through resonant interactions, as the optical field concentration generated by gold nanocrystals leads to various light–matter interactions, such as photothermal conversion and photochemical reactions. Focusing radiation on AuNPs at a frequency close to their resonant frequency enables them to convert that light into thermal energy. By controlling the size and shape of gold nanorods, their LSPR properties can be adjusted to exhibit two plasmon modes: longitudinal and transverse. By varying the length-to-diameter ratios, the plasmon wavelength of gold nanorods can be tuned across a broad range, including the visible and near-infrared (NIR) regions. These features make AuNPs suitable for light- or NIR-controlled drug release in drug delivery applications [27]. The most suitable AuNPs for such applications are gold nanoshells, nanorods, and nanocages, as they enable the adjustment of their physiological properties [28].

In comparison to other NPs used in therapeutic applications, AuNPs offer unique advantages. For example, while silver nanoparticles demonstrate strong antimicrobial and anticancer properties, they can release large amounts of ions rapidly, posing significant health risks. This can result in respiratory blockage, the generation of reactive oxygen species, and the inhibition of ATP production [29]. On the other hand, AuNPs are inert and do not react with the body’s internal environment. Additionally, quantum dots are well known for their excellent imaging applications due to their luminescent properties; however, they contain toxic metals that limit their therapeutic applications [30]. In contrast, AuNPs provide similar optical properties through LSPR while being safer and more biocompatible [26]. Moreover, iron nanoparticles cannot be surface-functionalized as easily as AuNPs due to their surface properties. Because of their high energy and surface chemical activity, iron oxide nanoparticles are easily oxidized, resulting in a loss of magnetism [31]. In contrast, AuNPs possess a high surface area and functional versatility, which prevent coagulation and enable precise targeting, biocompatibility, and greater control over drug release [25,32].

## 3. Different AuNP Shapes

The various characteristics and applications of AuNPs arise from their production in different shapes and sizes. Critical factors such as size and shape significantly influence AuNPs’ cellular interactions and their applications in the drug delivery domain. Additionally, these properties determine the safety and toxicity of the particles as they interact with cells and barriers within the body [33]. Different shapes of AuNPs include nanospheres, nanorods, nanocages, nanoshells, nanostars, nanotriangles, nanoclusters, and several others. The shape of the NPs is the primary factor influencing cellular uptake levels [34]. Although the various shapes of AuNPs can influence their properties and wide range of applications, such as biosensing and bioimaging, this paper will focus solely on the effect of shape on drug delivery and therapeutics.

Hollow gold nanospheres (HGNs) are highly valued in drug delivery applications due to their ability to load medications and their relatively lower toxicity compared to gold nanorods, which are synthesized using cetyltrimethylammonium bromide (CTAB), a toxic substance. HGNs are typically synthesized through the sacrificial galvanic replacement of cobalt NPs or a galvanic replacement reaction of silver for gold. These methods produce gold spherical NPs in the 20–50 nm size range, with tunable properties that enhance drug delivery and photothermal therapy. Additionally, surface modification improves biocompatibility and enables the active targeting of diseased cells. Active targeting involves attaching molecules such as antibodies and peptides to the surface of AuNPs to bind to tumor-specific receptors, facilitating effective cancer treatment. Photothermal therapy employs laser irradiation to raise the temperature of tumor tissue to lethal levels while sparing healthy cells. With their spherical shape and small size, these NPs are ideal candidates for photothermal therapy applications. Furthermore, HGNs loaded with drugs can combine the therapeutic effects of conjugation with targeting molecules and photothermal therapy to effectively kill cancer cells [35].

Gold nanorods (AuNRs) have unique therapeutic applications influenced by their formation mechanisms, including electrochemical synthesis, seed-mediated synthesis, and the ultraviolet photochemical reduction of gold salts, among others. Experiments have demonstrated that AuNRs are highly effective in cancer treatment, as they exhibit better permeation and retention in tumor tissue compared to normal tissue. This tumor retention of AuNRs is dependent on their aspect ratio. Although smaller AuNRs are cleared more rapidly, those with a high aspect ratio and small volume are ideal candidates for utilizing the EPR effect in tumor-mediated delivery [36]. Studies have shown that AuNPs with an aspect ratio of 3 to 5 exhibit the highest success in accumulating at the tumor site [36]. Another advantage is the high surface area-to-volume ratio, which provides more space for attaching recognition molecules, thereby enhancing targeting [37]. In addition, AuNRs have a longer circulation time compared to nanospheres, which are taken up by macrophages four times more efficiently [34]. Due to their anisotropic shape, AuNRs are characterized by two LSPR bands: the transverse band and the longitudinal band, which vary with the aspect ratio of the NPs [36]. Researchers have found that AuNRs emit stronger SERS signals compared to spherical AuNPs. This enhancement occurs when the wavelength of the excitation source matches the longitudinal wavelength of the nanorods, leading to more intense emitted SERS signals. Due to their outstanding photothermal conversion, AuNRs serve as excellent agents for the controlled release of drugs and tumor-killing applications [26]. Their tunable size and aspect ratio enable scientists to control their LSPR properties for thermal ablation therapy or photothermally controlled release [27].

Gold nanotriangles (AuNTs) are characterized by pointed extensions radiating from a spherical core and have been widely used in applications such as diagnosis, drug delivery, and imaging. The most popular synthesis method for AuNTs is seed-mediated growth, although reproducibility can be challenging. The longer side lengths of AuNTs are associated with a higher level of cellular uptake, attributed to the edges and vertices that have greater curvature. Studies have shown that AuNTs demonstrate higher cellular uptake than spherical AuNPs, with those having a side length of 50–150 nm exhibiting the highest cellular uptake due to their strong adhesion forces [38]. Cellular uptake consists of two steps, namely (1) the adhesion of NPs to the cell membrane and (2) the uptake of NPs into the cell through active mechanisms. The stability, biological compatibility, and high photothermal conversion of AuNTs make them excellent candidates for temperature-triggered drug release, as they have demonstrated promising results in cancer treatment research [34].

Gold nanostars are another type of AuNP that plays a pivotal role in drug delivery for cancer treatment. One study [39] synthesized gold nanostars with a diameter of 35.6 ± 3.6 nm through a seed-mediated route and functionalized them with a thiolated and carboxylated poly(ethylene glycol) (PEG) spacer, enabling covalent bonding with the amine-containing anticancer drug methotrexate (MTX). These nanoparticles possess unique optical properties that facilitate real-time imaging for drug delivery to cancer cells. Gold nanostars have significant potential for targeted therapy and precision medicine, as the study demonstrated their strong preference for accumulating within cancer cells and tumor tissues, particularly when encapsulating MTX. However, at high concentrations and with prolonged exposure, these nanoparticles exhibited cytotoxic effects [39]. Additionally, gold nanostars are among the most promising and widely used nanoparticles in cancer therapy due to their star-shaped geometry, which enhances light absorption and allows for highly efficient photon-to-heat conversion [18]. Their sharp structures at the tips, referred to as “hot spots”, significantly improve their photothermal properties by amplifying light intensity by up to 10^6^ times. Their “lightning rod” effect contributes to their strong surface-enhanced Raman scattering (SERS), which is greater than that of spherical AuNPs. Unfortunately, AuNTs also have disadvantages, such as low water solubility and a tendency to aggregate due to their sharp surfaces [40].

Gold nanoshells (AuNSs) are nanoparticles featuring an organic (polymer or lipid) or inorganic (metal) core coated with a thin layer of gold. They are widely used in drug delivery applications due to their unique properties. First, they exhibit resonance optical properties that depend on their size and shape, enabling deep tissue penetration. Additionally, they efficiently absorb light and convert it to heat, which is utilized for photothermal ablation to kill cancer cells. AuNSs can also serve as controlled drug delivery vehicles that release drugs at targeted sites upon near-infrared (NIR) irradiation. High-precision targeting is facilitated by functionalizing these particles with specific ligands, allowing them to bind to the receptors of designated diseased cells. AuNSs are particularly suitable for passive drug delivery, as they tend to accumulate in tumors through the enhanced permeability and retention (EPR) effect. Furthermore, functionalization with polyethylene glycol enhances their circulation time and helps them evade immune clearance. Hollow AuNSs can encapsulate anticancer drugs such as DOX and release them through NIR irradiation. AuNSs achieve active drug delivery by conjugating with targeting ligands (e.g., antibodies or peptides) to enhance their specificity towards cancer or other targeted cells. Additionally, AuNSs can be employed in gene therapy by encapsulating nucleic acids, protecting them from degradation, and enabling precise release via NIR irradiation [41].

Gold nanocages are a novel class of nanoparticles, typically ranging in size from 20 to 50 nm, that are particularly effective for photothermal conversion. They are synthesized through a galvanic replacement reaction using a silver template and gold salt, resulting in a hollow structure composed of a gold–silver alloy with adjustable wall thickness. Due to the differing chemical potentials of the two metals, silver ions dissolve into the aqueous HAuCl₄ solution while a gold layer forms on the outer surface of the nanocage. With their hollow interior and porous walls, gold nanocages offer significant potential for drug encapsulation and delivery. A thermosensitive polymer coating enables precise control over pore opening and closing, facilitating the release of drugs or other compounds. Gold nanocages employ both passive and active drug delivery methods, leveraging the enhanced permeability and retention (EPR) effect, as well as receptor overexpression on tumor cells. Additionally, gold nanostructures can be tailored for surface modification, allowing for the attachment of targeting molecules that promote receptor-mediated endocytosis [42].

Figure 2 illustrates the different shapes of AuNPs, which play a significant role in cellular uptake efficacy. Various shapes encounter different energy barriers when interacting with membranes, affecting their internalization rates. Spherical particles are well studied, having been the focus of numerous investigations on internalization and membrane wrapping. The wrapping of spherical particles is continuous, requiring low adhesion energy and minimal membrane energy barriers, which facilitates easier uptake compared to nonspherical particles. Elongated particles, such as gold nanorods, can enter cells through two modes, namely the “submarine mode”, where the particle’s long axis is parallel to the cell’s surface, and the “rocket mode”, where the long axis is perpendicular to the cell’s surface. Particles with a high aspect ratio often experience suppressed uptake, as the wrapping process is not smooth or continuous. The complete wrapping of long particles necessitates high adhesion strength and presents significant energy barriers. Cubic nanoparticles require minimal membrane deformation for initial binding, making their uptake easier and less dependent on adhesion strength. However, due to their inhomogeneous curvature, cubes need very high adhesion strength when transitioning from a deep-wrapped state to a completely wrapped state, compared to spherical NPs. Ellipsoidal particles exhibit lower uptake efficiency than spherical particles because the high curvature at their tips creates additional energy barriers. This results in a frustrated endocytosis state, where they are partially wrapped but not fully internalized. Nanorods typically bind perpendicularly to the membrane to minimize deformation energy but may switch to a parallel orientation depending on their aspect ratio, edge sharpness, and blunt tips. Nanoparticles with high edge curvature and aspect ratio are more challenging to wrap completely due to increased energy barriers [43]. The various shapes of AuNPs play a crucial role in determining their applications in drug delivery and therapeutics. The diversity in the shape of each AuNP leads to distinct properties regarding cellular interactions, targeting specificity, and photothermal capabilities. HGNs demonstrate significant potential in photothermal therapy, while gold nanorods exhibit superior cellular uptake and thermal properties. Other types of AuNPs, such as gold nanotriangles, nanostars, nanoshells, and nanocages, also possess advantageous features for cell targeting and temperature-triggered drug release. Despite their vast applications in drug delivery, careful consideration of toxicity is essential for ensuring patient safety and well-being.

## 4. AuNPs’ Synthesis

There are generally two approaches to synthesizing AuNPs: “Top-down” and “Bottom-up”, as illustrated in Figure 3. In the top-down technique, AuNPs are derived from bulk material using various methods such as laser ablation, ion sputtering, irradiation, and aerosol technology. During this process, energy transforms the bulk material into powder, which is then further fragmented into smaller pieces. These fragments are subsequently organized into multilayers and monolayers, leading to the formation of nanoparticles. In the bottom-up approach, Au^3^⁺ is reduced to Au in two stages. In the first stage, a gold precursor, typically an aqueous gold salt solution, is reduced to AuNPs using a reducing agent such as citrate. In the second stage, the AuNPs are stabilized by a specific capping agent that prevents nanoparticle aggregation. [44]. To successfully use AuNPs for biological and therapeutic applications, they should be synthesized to have characteristics such as water solubility, size dispersion, appropriate morphology, and surface functionalities customized for their purpose [32].

### 4.1. Chemical Synthesis

The Turkevich method, first reported in 1951, was used to synthesize spherical AuNPs with sizes ranging from 1 to 2 nm [45]. This bottom-up process involves reducing Au^3^⁺ to Au^0^ using reducing agents such as citrate, amino acids, or UV light, followed by the use of stabilizing agents to maintain the stability of the AuNPs. Advancements in this basic method have dramatically increased the size range of AuNPs synthesized, now spanning from 16 to 147 nm. While this method is simple and reproducible, it is mainly limited to the synthesis of spherical AuNPs. Additionally, as the particle size exceeds 30 nm, the shape tends to deviate from a perfect sphere [44]. Another study [46] described the Turkevich method, where chloroauric acid is reduced using trisodium citrate dihydrate to create AuNPs ranging in size from 15 to 100 nm. A higher concentration of citrate produces smaller and more stable AuNPs, while lower concentrations result in larger particles and increased aggregation. In this method, citrate acts as both a reducing and stabilizing agent, generating negatively charged AuNPs. The loose binding of citrate allows for easy replacement with other functional groups.

The Brust method, first reported in 1994, utilizes a two-phase reaction strategy to synthesize spherical gold nanoparticles (AuNPs) with diameters ranging from 1.5 to 5.2 nm [46]. This method uses a phase transfer agent to move gold salt from an aqueous solution to an organic solvent, typically toluene. The gold salt is then reduced by a strong reducing agent, such as sodium borohydride, leading to the formation of AuNPs, which are stabilized in the organic phase. A notable color change from orange to brown occurs during this process. The method is advantageous because it enables the easy formulation of thermally and air-stable AuNPs and allows for size control. However, the biological applications of AuNPs synthesized through this method are limited, as they can only be used in organic solvents that are immiscible with water [44]. Nicol et al. [46] described the Brust–Schiffrin method as a technique for synthesizing thiolate-stabilized AuNPs smaller than 5 nm, which exhibit high thermal stability. This method utilizes a strong reducing agent, sodium borohydride, resulting in smaller AuNPs compared to those synthesized using the Turkevich method.

The most common method for synthesizing gold nanorods is seed-mediated growth. This process begins with the synthesis of seed particles by using reducing agents to reduce gold salts. The seed particles are then transferred to a metal salt solution, where a reducing agent inhibits further nucleation and accelerates the formation of gold nanorods [47]. The concentration of reducing agents and seeds determines the shape and geometry of the nanorods. Additionally, the temperature must be carefully regulated, as it influences both the size and aspect ratio of the nanorods [44].

Digestive ripening involves heating a gold colloidal suspension at a high temperature of 138 °C for 2 min, followed by treatment with alkanethiol at 110 °C for 5 h. The size distribution of the gold colloids strongly depends on the heating temperature. It has been observed that at 60 °C, the particles remain stable at 5 nm. However, when the temperature increases to 120 °C, larger particles with a diameter of 7 nm are formed. Similarly, the particles grow to 10 nm at 150 °C and continue to grow rapidly at 180 °C [48]. This method is advantageous because it produces a high yield of nanoparticles. However, controlling the shape of these nanoparticles is challenging due to the high temperatures involved [44].

### 4.2. Biological Synthesis

Chemical methods have limitations, particularly regarding biocompatibility, which restricts their use in biological applications. Some of the chemicals used in synthesizing AuNPs can have harmful effects on the environment and pose health risks when introduced into living organisms. Consequently, biological techniques for synthesizing AuNPs are being developed, as they offer cleaner, more environmentally friendly, and reliable alternatives [44]. Microorganisms have been shown to be excellent agents for synthesizing both extracellular and intracellular AuNPs. The negatively charged bacterial cell wall can electrostatically interact with positively charged Au (III) ions. In intracellular synthesis, gold ions are transported into the bacterial cell, where enzymes and biomolecules facilitate the synthesis of AuNPs. In extracellular synthesis, gold ions are trapped by membranes or reductase enzymes on the bacterial cell surface, which convert gold salts into metallic AuNPs. Extracellular synthesis is preferred because it eliminates the need for an additional step to separate AuNPs from the intracellular matrix [49]. Proteins, enzymes, and other organic substances produced by microbial cells serve as stabilizing agents for AuNPs, preventing their agglomeration. Additionally, modifying specific growth parameters enables control over the shape and size of the AuNPs. However, the bacterial synthesis method can be tedious and time-consuming, which limits its application for producing AuNPs [44]. Fungi have been utilized as a source for the biological synthesis of AuNPs due to their ability to release various biomolecules, such as metabolites and enzymes. These agents facilitate the synthesis of metal nanoparticles, including AuNPs, using both unicellular and multicellular fungi [50]. Similar to the bacterial method, AuNPs can be synthesized either intracellularly or extracellularly using fungi. Fungi produce a variety of proteins and reactive compounds, making the synthesis process significantly faster than that with bacteria. However, genetically modifying prokaryotes to produce specific desired proteins can be challenging [44].

Synthesizing AuNPs from plants or plant extracts is an environmentally friendly, cost-effective, and rapid approach. This method utilizes harmless and biocompatible components, such as flavonoids, phytosterols, and quinones, to reduce and stabilize AuNPs [51]. Every part of the plant, including roots, fruits, and stems, can synthesize AuNPs; however, leaves are preferred due to their high phenolic content, which facilitates a faster synthesis process. Variations in compound levels among different plants and their parts can influence the synthesis of AuNPs. Using plants for AuNP synthesis is efficient and practical, allowing for the regulation of attributes such as shape and size. Nevertheless, identifying the reactive components in plants can be challenging, as they contain a wide array of bio-compounds [44]. Additionally, several species of algae can synthesize AuNPs, as algal biomass contains hydroxyl and carbonyl groups. These compounds serve as reducing and capping agents for AuNPs [52]. Although the synthesis of AuNPs using algae can be relatively simple and straightforward, the long growth duration of algae can make the process tedious [44].

Biomolecules such as amino acids, proteins, lipids, and carbohydrates contain various functional groups that facilitate the synthesis of AuNPs. For instance, chitosan is employed in the synthesis process because it can function as both a reducing and capping agent. Additionally, starch can be decomposed into carboxyl groups in an alkaline environment, where the -OH group of the carboxylic acid in starch can reduce Au^3^⁺ ions to Au^0^, leading to the formation of AuNPs. It is important to note that different biomolecules exhibit varying reducing abilities, which should be assessed before use [44]. The biological synthesis methods of AuNPs effectively address the biosafety concerns associated with the chemicals used in traditional chemical synthesis methods. Figure 4 and Table 2 provide a summary of the various synthesis methods, along with a discussion of the advantages and disadvantages of each approach.

## 5. AuNP Surface Functionalization

In drug delivery applications, it is essential to modify the surface of AuNPs to tailor them for targeting specific diseased cells, reducing off-target effects, and enhancing the drug’s therapeutic efficacy. Active targeting is achieved by attaching ligands, such as antibodies or other targeting agents, to the nanoparticle surface, as shown in Figure 5. One part of the ligand binds to the nanoparticle, while the other interacts with specific biomolecules on the target cells [32]. Functionalized AuNPs must remain in circulation for an extended period to effectively target diseased cells, such as tumors and sites of inflammation. Additionally, properties such as water solubility, the ability to evade the immune system, and resistance to hepatic and/or renal clearance must all be integrated into the same nanostructure to ensure successful tumor targeting [25]. This is achieved by attaching polyethylene glycol (PEG) to the surface of AuNPs, which prolongs their circulation time by preventing uptake and removal by macrophages [32]. Targeting functional groups or moieties can include antibodies, antibody fragments, aptamers, proteins, peptides, or small molecules [25].

Goddard et al. [53] discussed the various surface ligands on AuNPs, their purposes, and provided examples of each. Antibodies are predominantly used to target cancerous tumors due to their high affinity for specific receptors on cancer cells. For instance, trastuzumab and cetuximab are clinically approved antibodies that target the overexpressed HER2 and EGFR receptors, respectively. Antibody-functionalized AuNPs loaded with chemotherapy drugs are internalized via receptor-mediated endocytosis. Antibody fragments are created by reducing disulfide bonds, resulting in free thiols that can bind to the cores of AuNPs. These fragments exhibit cytotoxicity to cancer cells when exposed to radiation. Proteins can also be utilized to direct AuNPs, although they are less explored than antibody-targeted therapies. This is because many proteins chosen as targeting ligands are natural ligands in organisms. Their abundance in the human body leads to numerous competing ligands that are not associated with the AuNPs, thereby reducing targeting efficiency. Nevertheless, exploring the use of human proteins in targeted therapy remains valuable. For example, epidermal growth factor (EGF) is a protein that can be attached to the surface of AuNPs to target breast cancer cells that overexpress EGFR receptors. Another example is transferrin, a protein used to direct doxorubicin-loaded AuNPs to lung cancer cells.

Another commonly used targeting ligand is peptides, which are short polymers of amino acids utilized as targeting moieties for AuNPs due to their simplicity and rapid uptake. Although they exhibit lower affinity for receptors compared to antibodies, they offer significant flexibility. For example, the plectin-1 targeting peptide (KTLLPTPYC) is employed to direct AuNPs to specific targets, such as pancreatic ductal adenocarcinoma (PDAC). Additionally, peptides like GE11 on AuNPs loaded with photosensitizers target EGFR on glioblastoma cells and demonstrate significant phototoxicity. Peptides have advantages over antibodies and proteins in terms of their well-characterized structures, relative stability, and size. Moreover, they can be easily modified due to their synthetic production. Aptamers, which are short RNA or single-stranded DNA sequences, selectively bind to receptors, enabling precise NP targeting. Aptamers provide advantages such as high precision in synthesis, versatility across various applications, high stability, consistency, and reproducibility, since they are synthetically produced rather than naturally derived. They can also be designed to target multiple receptors, potentially creating a synergistic effect when combined with other targeting ligands or drugs. Additionally, carbohydrates such as hyaluronic acid, lactose, and glucose are employed to direct AuNPs toward cells with overexpressed receptors. Hyaluronic acid typically targets CD44 receptors, while glucose targets GLUT1 receptors, both of which are often overexpressed in cancer cells. AuNPs can also be directed by small molecules such as folic acid, anisamide, and antiandrogens, which provide a promising approach for targeted cancer therapy due to their high affinity for receptors on tumor cells.

There are two methods for conjugating targeting moieties onto functionalized AuNPs: noncovalent and covalent conjugation. Noncovalent attachment methods include weak interactions such as electrostatic interactions, hydrogen bonds, van der Waals forces, and hydrophobic interactions, which are used to link targeting moieties to the surface of AuNPs. This method is advantageous because it is simple and applicable to various types of moieties or drugs. However, these loose attachments may lead to molecular rearrangements and changes, resulting in a loss of biological properties. In contrast, covalent bonds are stronger and typically involve Au-S bonds formed through thiol chemistry.

Antibodies, antibody fragments, aptamers, proteins, peptides, or small molecules can be covalently bonded to the surface AuNPs. The modifiable groups (-NH_2_, -SH, -COOH, -OH) on their surface enable covalent bonding. Compared to noncovalent methods, strong covalent bonds reduce molecular rearrangement and, in turn, allow the AuNPs to retain their biological properties and targeting abilities [25].

The hetero-bifunctional PEG molecule was synthesized with a thiol group on one end and an alcohol carbamate on the other end, which allowed coumarin to attach to PEG [54] covalently. Coumarin was used in this context as a fluorescent dye to track AuNPs within biological systems. Using PEG spacers offered flexibility to the attached molecule, enhancing its interaction with the biological targets and preventing the aggregation of AuNPs through steric repulsion. The same study proved that AuNPs functionalized with coumarin-PEG-thiol were not toxic at the concentrations tested. It observed that, within 30 min of incubation, large portions of the administered dose of coumarin-PEG-thiol-functionalized AuNPs were internalized via nonspecific endocytosis. It was also proven that the hetero-bifunctional PEC provided flexibility to the florescent ligand without causing aggregation and hindering the AuNPs’ entry to the perinuclear region.

The functionalization of citrate-capped AuNPs is straightforward due to the weak electrostatic interactions between gold and citrate, which can be displaced by stronger thiolated ligands such as polymers, fluorescing dyes, drugs, and peptides. These thiol-terminated ligands replace the more loosely bound citrate or phosphine on the AuNP surface. Additionally, the Turkevich method produces AuNPs with a negatively charged citrate capping, facilitating electrostatic functionalization with positively charged ligands. Unfortunately, this method suffers limitations, such as the need for positively charged ligands and challenges in controlling biological responses. To create AuNPs for therapeutic use, it is necessary to functionalize them with several ligands, attaching them to the surface as a mixed monolayer. The standard mixtures of such a monolayer include combinations PEG and peptides, PEG and fluorescing dye, or PEG with heavy metal atoms [46].

The R. W. Murray research group [54] found that triphenylphosphine (TPP)-stabilized AuNPs undergo reactions through which they exchange their ligands with certain functionalized thiols, resulting in functionalized NPs. These particles retained their core properties while gaining advantages such as stability against heat, decomposition, and aggregation due to functionalization. The intense interaction between the gold surface and n-alkyl-thiols is a widely used method for attaching molecules to the surface for functionalization. One limitation of this study is that this bonding is reversible at moderate temperatures and kinetically unstable due to thiol surface movements. As a result, it is challenging to precisely fixate functional groups on metal surfaces and control their activity on an angstrom level. This can be achieved by knowing the building blocks’ structures at the atomic level [37]. Another limitation is the diminishing colloidal stability of colloidal AuNPs as their size decreases. This causes them to aggregate in environments with high ionic strengths. This leads to the unspecific adsorption of biomolecules such as proteins and DNA, reducing the sensitivity and selectivity of the colloidal AuNPs [32].

Even though various materials and techniques are available to synthesize AuNPs, customizing them to therapeutic and biological applications has limited their synthesis to a few chemical methods. AuNPs tailored for drug delivery must be biocompatible and functionalized appropriately based on their purpose and target cells. In addition, their core characteristics must be retained while enhancing the NPs’ stability against heat, aggregation, and biological conditions. There is an increasing demand for more straightforward methods to attach various ligand molecules to AuNPs’ surfaces while retaining their stability properties.

### Surface Functionalization and Biocompatibility

To synthesize AuNPs for therapeutic applications, they must be biocompatible to avoid cytotoxicity. According to Sanita et al. [55], several strategies focused on the surface functionalization of NPs can be utilized to achieve biocompatibility. PEGs are widely used molecules to achieve biocompatibility and reduce toxicity. This is achieved by improving hemocompatibility, increasing cell viability and reducing adverse effects in the blood. In addition, their presence on the surface reduces the risk of immune recognition and clearance by the liver and kidneys by preventing nonspecific interactions with proteins. A complex branched polysaccharide called Dextran and an oligosaccharide called chitosan are other modifying materials used to enhance biocompatibility, reduce toxicity, and minimize effects on cellular integrity. Other complex methods include the lipid bilayer encapsulation of the NPs, which improves biocompatibility, reduces toxicity, and enhances stability by reducing the immune response towards them. In addition, coating NPs with a hydrophilic coating makes them less viable to the immune system, reducing interaction with macrophages. This increases their circulation time and reduces their clearance.

Transcytosis, a process by which membrane-bound vesicles transport molecules across endothelial cells, aids in transporting AuNPs across body barriers. A major plasma protein called albumin interacts with AuNPs and coats them to facilitate their transport through transcytosis, enabled by receptors on endothelial cells [56]. Differences in the structure and function of endothelial cells in different organs influence the transport and accumulation of NPs. As a result, AuNPs must be functionalized appropriately to show association with blood proteins and achieve organ-specific drug targeting. For example, hepatocytes, which are the main cells of the liver, offer an advantage for NP-targeted delivery; this is due to the surface-engineered targeting molecules on the AuNPs’ surface, such as alpha-tocopherol (vitamin E) or cholesterol [57]. Figure 6 summarizes the different purposes of surface functionalization of AuNPs, which is essential for enhancing their stability, biocompatibility, targeting capabilities, and other characteristics in biomedical applications.

## 6. Drug Loading onto AuNPs

Understanding the mechanism of drug encapsulation in AuNPs is highly important for designing and optimizing their therapeutic effect. This section discusses a few case studies describing the different methods to load a drug into AuNPs, including encapsulation or surface attachment. Different conjugation methods are used to attach drug loads to AuNPs. Ionic interactions happen when positively charged drug loads are attracted to negatively charged AuNPs’ surfaces. Another method utilizes the covalent bonds that bind conducting electrons of gold and the sulfur of the drug load. Also, the interaction between the hydrophobic region of the AuNPs’ surface and that of the drug promotes binding. Another method is chemisorption, which is due to thiol derivatives enhancing the attachments of drugs on the surface of the AuNPs. An alternative method is when bi-functional linkers such as biotin and streptavidin enable the attachment of drugs to AuNPs through specific binding interactions [17].

Immunotherapy is used to reactivate the body’s immune response to combat cancer using immunomodulators such as cytokines, vaccines, inhibitors, and adjuvants. If used directly in cancer immunotherapy, immunomodulators suffer disadvantages such as drug instability, rapid clearance, limited half-life, and unpredictable immune response. As a result, this drug must be loaded into AuNPs, as studied in [18], as these NPs are biocompatible and administered intravenously, increasing cell permeability and retention. This study reviews the recent methods of encapsulating immunomodulators of AuNPs for cancer treatment. It claims that encapsulated AuNPs can maintain drug concentrations within the therapeutic range, provide the precise dosage at the optimal time and location in the body, and use the drug effectively. On the other hand, non-encapsulated drugs can cause concentration fluctuations in the blood, cause uneven drug distribution, and cause side effects in the body as they target normal, healthy cells. Various drugs can be encapsulated or loaded onto AuNPs through binding methods such as direct -S (sulfur) or -N (nitrogen) bonding, ligand bonding, van der Waals forces, or electrostatic interaction. According to the study, nitrogen binding has more potential for targeted drug delivery than sulfur binding due to the strong sulfur–gold interaction.

Vigderman et al. [58] described two drug-loading mechanisms for AuNPs’ drug delivery applications, namely bactericidal and anticancer. In an antibacterial implementation, antibiotics containing free amino groups are bonded to AuNPs through covalent bonding or by mixing them with citrate-capped AuNPs. One limitation of this method is the possible aggregation of AuNPs, which reduces bacterial activity. As a result, the NPs should be functionalized to prevent aggregation. In addition, it is problematic to mix AuNPs with molecules containing multiple amine groups, as the gold–amine interaction is weak and could cause the uncontrolled aggregation of particles. Another type of antibiotic loading is the covalent gold–thiol binding approach. This method involves coating AuNPs with antibiotics through direct covalent bonding or phase transfer mechanisms. Covalent bonding is advantageous as its strength ensures the stable attachment and precise orientation of the antibiotic on the outer surface of the AuNPs. Covalent bonding is also beneficial, as it provides specific bonding to targeted cells, reducing side effects. The same study discusses that the conjugation of anticancer drugs is also carried out through covalent bonding, taking advantage of the AuNPs’ established chemistry.

Venditti et al. [15] investigated the drug-loading process and properties of dexamethasone (DXM), a glucocorticoid used to treat myeloma and various inflammatory diseases. This drug arose as a prominent candidate for AuNPs’ encapsulation, as its hydrophobic nature makes it suitable for tissue engineering applications. The study focuses on the electrical interface of AuNPs’ bioconjugation with DXM and evaluates its drug-loading capacity. They synthesized spherical AuNPs through chemical reduction methods using sodium borohydride as a reducing agent and Sodium-3-mercapto-1-propansulfonate (3MPS) as a stabilizing agent, which is crucial for biomedical applications. The protocol for loading DXM into AuNPs resulted from their interaction in a dispersing solution at room temperature with a weight ratio of AuNPs 5:1 DXM, resulting in the adsorption of the drug into the AuNPs.

Tan et al. [59] discussed the incorporation of cisplatin (PtII) in their study, a drug that treats various solid tumors and inhibits their replication, in AuNPs. The study found that PtII loading increases with concentration until it reaches a plateau, indicating saturation. The chemotherapy drug is incorporated onto AuNPs by binding to free carboxyl sites on the surface of the NPs. In addition, the researchers studied the effect of SH-PEG (thiol-modified PEG group) on PtII loading. They found that it is insensitive to the surface group density if the drug has access to enough available carboxyl sites.

## 7. AuNPs’ Targeting Mechanisms

AuNPs target diseased cells in different ways, including active and passive targeting. Endocytosis is a complex cellular uptake mechanism that depends on several factors. Chemically modified AuNPs can reach their target, penetrate barriers, and avoid degradation by coating them with cell-penetrating proteins or nuclear-localizing agents. If AuNPs were administered intravenously, they must possess characteristics that allow them to move from blood vessels to surrounding tissues or, if administered intramuscularly or subcutaneously, diffuse easily through tissues [60]. NPs’ characteristics, such as shape and surface charge, affect their interaction with cell membranes, affecting delivery, uptake, or permeability. Surface modifications such as functionalization, PEGylation, and other physiochemical properties, such as size and shape, affect the targeting abilities of AuNPs and their internalization into cells [17].

### 7.1. Active Targeting

Active targeting is when AuNPs use target ligands or moieties that bind to specific receptors on targeted cells, enhancing specific drug uptake, as shown in Figure 7. Such ligands are attached to the surface of NPs through conjugation chemistry to bind to complementary biomarkers on the surface of diseased cells, leading to highly specific targeting. Once bound, NPs are internalized into these cells through receptor-mediated endocytosis [17]. Serum proteins, such as albumin, fibrinogen, and globulins, interact with AuNPs and bind to them through electrostatic interactions. Such proteins coat AuNPs and allow their uptake by cells through receptor-mediated endocytosis. AuNPs’ specificity increases by coating their surface with ligands that bind to known receptors or by using passivation/blocking agents to block the adsorption of nonspecific serum proteins. Coating with serum proteins may be advantageous or compromise targeted delivery, so careful consideration must be given to specific targeting. For example, deciding whether specific uptake processes or nonspecific methods are needed for an application to coat the AuNPs with targeting molecules is crucial. In addition, there might be possible changes in the AuNPs’ conformation, or three-dimensional shape, due to binding to peptides or antibodies [60]. Active delivery is highly utilized when delivering macromolecules such as DNA, siRNA, or proteins into the cells [17].

As described by Bareford et al., clathrin-mediated endocytosis is the main mechanism used to internalize macromolecules such as drug-loaded NPs. [61]. Endocytosis is initiated when specific ligands on the NP attach to the appropriate receptors on the targeted cells’ surface. Then, coated pits with clathrin proteins form the cell membrane in a lattice structure. These pits start to undergo rapid internal budding, forming intracellular vesicles and internalizing the bound NPs. Dynamin protein then assists in the scission of the vesicle from the plasma membrane, so the vesicle gets released into the cytoplasm. After that, the clathrin coat is removed by uncoating proteins as the vesicle is transported along the cytoskeleton to fuse with early endosomes to process the internalized components. Early endosomes increase the acidity within the vesicles to trigger the release of bound substances, which are further transferred to interest sites in the cell. Targeted drug delivery utilizes clathrin-mediated endocytosis to internalize drugs into various cell types with the appropriate surface receptors.

### 7.2. Magnetic Targeting

AuNPs can also be utilized for magnetic delivery, as studied by Ma et al. [62], where CDF-Au-shell nano micelles loaded with doxorubicin, Fe_3_O_2_ magnetic nanoparticles, and gold nanoshells were synthesized. These nanomicelles make a smart nanosystem for drug delivery that can be tracked by magnetic resonance imaging (MRI), delivered by a magnetic field, and its drugs released by light. A synergistic effect between magnetic-field targeted delivery and photothermal therapy in the presence of NIR laser-irradiation-treated cancer cells. The average size of these particles was 273 nm, with an average surface charge of 23.8 mV. Then, negatively charged gold nanoseeds were adsorbed to the micelles’ surface, resulting in an average surface charge of −13.8 mV, demonstrating a strong response to a magnetic field. Another study by Elbaily et al. [63] involved the loading of doxorubicin on PEGylated magnetic AuNPs. Using an external magnetic field, magnetically targeted delivery resulted in a high drug accumulation at the tumor site compared to passive targeting, as proved in in vivo studies. In this study, the positively charged DOX molecules were electrostatically attached to the AuNPs’ negatively charged surface due to charged carboxyl groups.

### 7.3. Passive Targeting

Passive targeting utilizes the enhanced permeability and retention (EPR) effect in tumor vasculature to target cancer cells. This is because tumors have poor lymphatic drainage, preventing the efficient removal of foreign NPs and highly permeable blood vessels. Microvasculature in tumor sites lacks membrane support, which makes it unresponsive to triggers that regulate blood flow [64]. This abnormal physiological characteristic, such as disorganized blood vessels, allows NPs to passively accumulate at the tumor site. Its limitation is nonspecific accumulation in healthy tissue and the heterogeneity of the EPR effect in tumor sites [17].

According to Ejigah et al.’s [64] review, successful passive targeting must meet several criteria, as shown in Table 3. First, NPs must be in systemic circulation long enough to accumulate in the tumor site. Second, the NP drug carrier must be small enough to escape blood vessels and fit into the spaces of the irregular tumor shape. Also, the surface charge and chemistry should be appropriately designed to penetrate tissues and avoid opsonin aggregation, enhancing phagocytosis and marking the foreign NPs for recognition by immune cells. This can be achieved by having a uniform size distribution to avoid recognition and elimination by the reticuloendothelial system. Finally, nonspecific release should be avoided to prevent side effects.

## 8. AuNPs’ Drug Release and Therapy Mechanisms

AuNPs release their drug content to target cells or tissues in the body using various mechanisms tailored for specific purposes and conditions. Controlled drug delivery can be carried out by AuNPs through intrinsic and extrinsic triggers. Intrinsic triggers include biological stimuli like pH, redox reactions, and enzyme activity, which trigger the release of drugs from the AuNPs’ surface in response to internal body conditions. For example, pH-responsive AuNPs release doxorubicin in acidic tumor environments, and redox-responsive AuNPs release drugs in the presence of high glutathione (GSH) levels. Extrinsic triggers, such as light and ultrasound (US), control drug release externally. For example, light-controlled methods utilize NIR light to heat AuNPs, causing drug release and localized tumor ablation. In addition, US triggers release drugs by causing cavitation near AuNPs, disrupting the bonds attaching drugs to the surface of AuNPs and allowing for their release into the surrounding cells.

### 8.1. Light-Controlled Release and Photothermal Therapy

As discussed in the Properties section of this paper, the plasmonic properties of AuNPs make them uniquely suitable for light- or NIR-controlled release of drugs at target sites in the body [27]. A study by Huschka et al. [28] investigated the light-triggered release of single-stranded DNA from plasmonically tunable gold nanoshells and compared the light-induced and thermal-induced release differences. First, gold nanoshells were functionalized with double-stranded DNA oligonucleotides. The thermal treatment of gold nanoshells included heating the solution; meanwhile, the laser treatment involved irradiating it with an NIR laser at the nanoshells’ plasmon resonant frequency. Localized heating due to laser irradiation causes stress between the DNA molecules and the surface of the AuNP, weakening the bonds between them and causing the release. Thermal release relies on the elevated temperature overcoming the affinity between DNA molecules and the AuNP’s surface, promoting their desorption and release. In both cases, the elevated temperature breaks the hydrogen bonds holding the double-stranded DNAs’ complementary strands together, causing the release of single-stranded DNA. The results demonstrated that NIR-controlled release was more immediate and highly reproducible within each batch compared to thermal release.

Another study by Veeren et al. [65] reviewed the NIR-triggered release of contents in liposomes with hollow gold nanoshells on their outer surface. Liposomes have a physical bilayer that separates their content from the outer environment and protects them from degradation. Attaching hollow gold nanoshells to their exterior allows for the NIR-controlled release of the liposome-encapsulated drugs, as the gold coating provides a tunable LSPR. The mechanism involves picoseconds to nanoseconds of NIR laser rapidly heating the AuNPs on the surface of liposomes, forming vapor nanobubbles near the surface. As these bubbles collapse, mechanical forces strong enough to rupture the liposomal membrane are generated, causing the release of contents at the selected time and site in the body [65].

Photothermal therapy (PTT) is when AuNPs inside diseased cells get heated up in response to externally applied laser light. Due to their plasmonic resonance properties, AuNPs can convert laser light to thermal energy. Thus, using them as thermal agents, AuNPs can heat tumor sites, causing damage to them or ablating them directly. Light-controlled release and thermal ablation can simultaneously create a synergistic effect in killing cancer cells [66]. PTT is different from light-controlled release, as it kills the target cells by heating the AuNPs and causing thermal ablation instead of releasing the drug, because heat energy breaks bonds.

### 8.2. Drug Release Based on Biological Stimuli: pH, Redox Reaction, and Enzyme Activity

AuNPs that can release drugs based on a trigger, such as a change in pH, temperature, and redox environments, are dependable because they can simply be triggered by pathological events specific to the targeted cells or area in the body. For example, tumor tissue is characterized by a lower pH, high intercellular GSH, tumor-specific enzymes, and slightly higher temperatures compared to healthy tissues [67].

One study by Khutale et al. [68] synthesized a pH-responsive PEGylated drug-conjugated AuNPs covalently coupled with polyamidoamine (PAMAM) G4 dendrimer. The conjugated drug doxorubicin (Dox) is used in cancer therapy, and its release in the acidic compartment of cancer tumor sites was tracked using confocal laser scanning microscopy (CLSM). The study reported negligible release at physiological pH and approximately 50% release at acidic pH after 96 h. The low pH breaks the amide bond between the drug (DOX) and the dendrimer (PAMAM), releasing the drug. The study also compared the cytotoxicity of Au-PEG-PAMAM-DOX, free DOX, and Au-PEG-PAMAM nanoparticles and found that Au-PEG-PAMAM-DOX was the most effective group, indicating high efficiency in killing cancer cells. CLSM confirmed that the AuNP system released DOX at the lysosomal sites of cancer cells with an acidic pH. The drug-release kinetics followed a zero-order model, where the controlled drug release did not depend on its concentration in the system and occurred at a constant rate.

Another study by McIntosh [69] explored the effect of GSH, a chemical abundant in tumor sites, as a biological trigger to release the contents of AuNPs in tumor environments. As AuNPs encounter an environment with high GSH levels, these GSH molecules interact with the positively charged surface of the AuNPs through redox disulfide exchange reactions. This process reduces the charge of the AuNPs surface, leading to a decrease in the affinity of the payload adsorbed on them and, eventually, their release.

A study by Schneider et al. [70] studied the release of AuNPs functionalized with a terpolymer of three different monomers due to enzymic activity. The outer monomer (Ma-Y-Dox) contains a form of DOX that should be released to the tumor site. A spacer between the drug and the rest of the oligopeptide is designed to be broken upon exposure to the lysosomal enzyme Cathepsin. This exposure will lead to the release of Dox exclusively within the environment where Cathepsin is present.

### 8.3. Photochemical Release

Within photochemical systems, an encapsulated prodrug is covalently bonded to an AuNP, where its activity is inhibited as it is attached to a blocking molecule through a photo-responsive group. Upon the exposure of AuNPs to UV light between 250 and 380 nm, the light interacts with the photoactive part of the molecule and releases the active form of the drug [67]. For example, Nakanishi et al. [71] used this approach to deliver amines released after exposure to 10–1000 ms pulses of near-UV laser irradiation at 365 nm with a 100 mW/cm^2^ intensity. This irradiation dissociated a carbamate linkage by photocleavage, after which the histamines were released and activated.

### 8.4. Ultrasound Release

Ultrasound can also be used to release the drug loaded on the surface of AuNPs to the targeted area in the body. In a study by Jakhmola et al. [72], anti-cancer drugs, doxorubicin, and curcumin, were loaded onto the surface of the AuNPs during their formation process, a one-pot green protocol. A custom-designed handheld low-power-intensity ultrasound (LIPUS) [73] device was utilized for an ex vivo drug release experiment. The transducer has a central frequency of 1 MHz, a pulse repetition frequency of 1 kHz, a fixed exposure time of 5 min, and a maximum power of 8.40 W ± 1.4%. In addition, it can operate on three power intensities, low, medium, and high, and on three different duty cycles, 40%, 50%, and 100%. The researchers aimed to raise an ex vivo porcine muscle tissue sample to reach hyperthermia without causing thermal ablation, so they used the LIPUS transducer with a 4.12 W power intensity and 50% duty cycle. This led to the synergistic effect between the effective release of drugs due to US and hyperthermia. US waves cause cavitation near the surface of AuNPs, where the turbulence disrupts the bonds between the drug molecules and the AuNPs’ surface to which they are attached. Another study by Yoon et al. [74] discussed applying US on microbubbles that released AuNPs to the target site. US above the resonance range of smart gold microbubbles can lead to their collapse of and release of AuNPs. This collapse will induce sonoporation in the cell membranes of the surrounding cells, as the released AuNPs can easily diffuse into them.

Yoon et al. [74] applied smart microbubbles filled with gas and shells, including AuNPs, to treat tumors using US for image-guided delivery. This type of delivery has been studied recently due to advancements in US contrast agents (UCAs), which are highly effective in diagnosing and treating tumors. UCAs consist of a gaseous core encapsulated in microbubbles that interact with US to disrupt the cell membranes of neighboring cells through shockwaves and facilitate the delivery of therapeutic agents into the cells. On average, the diameter of a microbubble was 1 μm and fit around 1.5 × 10^4^ AuNPs in it in this research. US above the resonance range of smart gold microbubbles can lead to their collapse and the release of AuNPs. The collapse will induce sonoporation in the cell membranes of the surrounding cells, as the released AuNPs can easily be delivered into the cells. The microbubbles were stabilized due to the aggregation of AuNPs, which resulted from the pH-responsive ligands on their surface due to the acidic microenvironment of tumor sites. The study found that a synergistic effect between these AuNP-filled microbubbles, US, and laser irradiation produced an effective therapy, as the laser-induced enough energy to be absorbed by AuNPs and heat the host–cell to kill it. Without this laser irradiation, no therapeutic effect was absorbed. Table 4 outlines the advantages and disadvantages of intrinsic and extrinsic drug release mechanisms for drugs loaded on AuNPs.

## 9. AuNPs Therapeutic Applications in Drug Delivery

### 9.1. Cancer Treatment

Recent advancements in accurately controlling their size, shape, and surface chemistry make gold nanoparticles excellent anticancer drug carriers. Adding a biocompatible surface coating to AuNPs renders them stable in physiological conditions. In addition, adding surface functionalization allows them to target specific receptors that are overexpressed on cancer cells. Ease of synthesis, chemical stability, and interesting optical properties have made researchers tremendously interested in utilizing AuNPs for cancer treatment. AuNPs’ role in cancer theragnostics, drug administration and loading, dose degradation, movement in the body, accumulation in the tumor site, and excretion is analyzed and understood by scientists [66].

AuNPs can target tumor sites through passive methods, such as the EPR effect, or active methods, such as ligand–receptor affinity. In addition, several cellular uptake methods, such as receptor-mediated endocytosis and sonoporation, are utilized. Through covalent or noncovalent bonds, AuNPs can either be loaded with anticancer drugs or encapsulated, as several release methods, as mentioned in this paper, separate the drug from the NP. PTT also kills tumor cells through hyperthermia while avoiding heating up surrounding healthy cells. Table 5 and Table 6 present some case study examples where AuNPs were used as therapeutic or drug delivery agents to treat cancer tumors, as revealed in modern research. For example, a case study by Lopes-Nunes et al. [77] researched the use of AS1411-AuNPs for treating cervical cancer. The AuNPs are covalently bonded to the AS1411 aptamers, enabling NPs to actively target cancer cells. In addition, their size of 18.3–12.16 in diameter makes the AuNPs small enough to utilize the EPR effect to accumulate in the tumor site. The drugs C8 or IQ are associated with the AuNPs via supramolecular assembly or noncovalent interactions on the surface of AS1411-AuNPs, which are internalized into cells through receptor-mediated endocytosis. Table 5 and Table 6 summarize different in vivo and in vitro research, where AuNPs were used as drug delivery or PTT agents for cancer treatment.

### 9.2. Skin Diseases Treatment

The skin serves as a natural barrier against drugs, especially those that are highly hydrophilic or have a heavy molecular weight. As a result, the unique properties of AuNPs, such as size, surface chemistry, and shape, have been investigated lately for skin applications, as researchers are finding ways to overcome the skin barrier and improve hydrophilic or macromolecular drug delivery onto or through the skin. A review by Chen et al. [95] summarized the factors that make AuNPs optimal for dermal or transdermal drug delivery, such as biocompatibility, low toxicity, and large surface area, and captured the drug loading, release, and penetration methods. Size affects the penetrability of AuNPs, as smaller NPs tend to penetrate the skin more effectively than larger ones. This is because smaller AuNPs have a higher surface area-to-volume ratio and greater mobility, which makes their interaction with the skin easier. Another factor that plays a role in skin penetration is surface chemistry. PEGylation is highly used, as it improves biocompatibility and stability and increases the penetration of the AuNPs into the epidermis, subcutaneous tissues, and hair follicles. In addition, PEG chains generate steric repulsions, which prevents the aggregation of the NPs. Also, the surface of AuNPs can be modified by targeting moieties such as cell-penetrating peptides and aptamers, enhancing their specificity and therapeutic outcome. Size is an important aspect to consider when aiming for high skin penetrability, as AuNPs with different aspect ratios interact differently with the skin. For example, gold nanorods penetrate the skin more efficiently than gold nanospheres because of their elongated shape, which facilitates intracellular penetration. Also, shape affects the surface area-to-volume ratio, which dictates the number of surface modifications and the surface chemistry of the AuNPs. As discussed in the AuNPs’ shapes section, the shape of the NP affects cellular uptake, as some shapes are more readily absorbed into cells than others. Lastly, different shapes can trigger different immune responses, so choosing an appropriate AuNP shape is crucial when considering dermal and transdermal drug delivery applications, as shape affects NPs’ interaction with skin and penetration abilities. In comparison to gold nanospheres, gold nanorods and nanostars are capable of greater deposition in the epidermis and subcutaneous tissue, due to their structure that increases friction and shear rates at the skin interface [95]. In addition, AuNPs with sharper shapes have higher photothermal activity, increasing their therapeutic effect, because their sharp ends have regions called ‘hot spots’ that significantly improve near field enhancement [40].

As shown in Figure 8, AuNPs penetrate the skin using different routes, including transcellular, intercellular, and transappendageal penetration. Transcellular penetration is the process by which AuNPs pass directly to the cell membrane through the cell cytoplasm to exit or enter it. This method requires the diffusion of drugs across the cells’ lipids bilayer, which is suitable for small lipophilic molecules that can dissolve in the cell membrane and reach the intracellular space. This process occurs through passive diffusion, facilitated diffusion, or active transport techniques depending on the physical and chemical properties of the drug molecule. Intercellular penetration involves the movement of drugs between the tight intercellular gaps and junctions without crossing cell membranes through diffusion or convection. This route is advantageous for hydrophilic drugs that cannot easily cross the cells’ lipid bilayer. The last method, transappendageal penetration, involves the movement of drug molecules into the skin through appendages such as hair follicles, sweat glands, or sebaceous glands. These skin appendages provide a route for drugs to bypass the corneum, the main skin barrier. The size, density, and activity of skin appendages dictate the efficiency of transappendageal penetration [95].

### 9.3. Ocular Diseases Treatment

Ocular drug delivery is a promising route to deliver drugs used for the treatment of several ocular diseases, such as diabetic retinopathy, which is triggered by inflammation and increased levels of vascular endothelial growth factor (VEGF) or neovascularization. According to Apaolaza et al. [96], AuNPs can inhibit angiogenic molecules, as they have unique antioxidant and antiangiogenic properties that make them suitable for ocular drug delivery. Also, their ability to conjugate a wide variety of biomolecules and surface functionalization can enhance the AuNPs’ movement through physiological eye barriers such as virtuous humor. The researchers employed hyaluronan (HA) on the surface of AuNPs to increase and facilitate their mobility across barriers and enhance their targeting abilities in detecting HA receptors in different eye cells. The study demonstrated that the conjugation of AuNPs with HA increased their stability, distribution, and mobility through the vitreous humor and deep retinal layers.

The HA group was attached to the surface of AuNPs through thiolation, the attachment of the thiol (-SH) group into the HA polymer chain. This modification allows the HA group to form a stable covalent bond with the surface of the AuNPs. The functionalization of HA-AuNPs prevented their aggregation in the cell cytoplasm with respect to AuNPs only. In addition, cellular uptake through receptor-mediated endocytosis differed between AuNPs and HA-AuNPs, as the latter possesses faster uptake since the HA causes specific interactions with CD44 receptors expressed on the eye cell surface [96].

### 9.4. Diabetes Treatment

Diabetes has contributed to a significant increase in health problems and death rates, as it causes long-term implications such as retinopathy, nephropathy, and peripheral neuropathy and could lead to organ failure due to uncontrolled high glucose blood levels. Alomari et al. [97] reviewed the different properties of AuNPs, such as their anti-hyperglycemic, antioxidant, anti-inflammatory, and antiangiogenic potential, that make them suitable candidates for the treatment of diabetes. One outcome of hyperglycemia is mitochondrial respiration, which releases reactive oxygen species (ROS). Diabetic complications result from oxidative stress generated by ROS, which are highly reactive molecules, or free radicals, that destroy cellular components. The antioxidant properties of AuNPs prevent ROS release to the cytoplasm, as they scavenge free radicals by increasing the activity of enzymes such as superoxide dismutase and glutathione peroxidase, which neutralize the activity of ROS. In addition, the surface chemistry of AuNPs allows them to donate electrons to free radicals, neutralizing their activity and reducing their concentration in the cell. The anti-hyperglycemic properties of AuNPs stem from the fact they can regulate blood sugar levels by promoting the generation of pancreatic β cells that produce insulin and by modulating glucose metabolism pathways. AuNPs also provide effective therapeutic effects for other diabetic complications, such as wound healing, effects on the retina, and neuroprotective problems. AuNPs can improve wound healing, as their anti-inflammatory effects reduce inflammation at the wound site and promote tissue regrowth. Also, they can have a therapeutic effect on diabetic retinopathy by inhibiting neovascularization, or the abnormal growth of vessels in the retina. This is carried out by AuNPs suppressing VEGFR-2 (vascular endothelial growth factor receptor 2), a receptor responsible for triggering neovascularization.

### 9.5. Antibacterial Effect of AuNPs

Antibiotic resistance is why new and more effective treatments against bacterial infections are being developed. Colistin is an antibiotic that is considered the last line of defense, but it has not been used much lately due to its undesirable side effects; therefore, it has reduced resistance. Recently, it has been considered as a last resort for treating multi-drug-resistant bacterial infections, as rising pathogens are becoming increasingly resistant to common antibiotics. Since Colistin resistance is dose-dependent, AuNPs are being investigated to deliver this antibiotic to treat infections with a lower dosage; yet, they have been shown to produce the same therapeutic effect by Fuller et al. [98]. Having a high surface area-to-volume ratio and low toxicity, Colistin-coated AuNPs create a system that delivers low antibiotic dosages, overcoming resistance issues. The antibiotic is attached to citrate-capped AuNPs through electrostatic self-assembly, resulting in enhanced effectiveness against the antibiotic with respect to the antibiotic alone and minimized side effects. In general, AuNPs can disrupt the bacterial cell wall by inducing mechanical stress. As AuNPs adhere to bacterial membranes in large amounts, they trigger membrane wrapping, increasing tensions and deformity in the cell wall. This causes mechanosensitive channels to open and creates an osmotic pressure inside the cell, leading to the membrane’s rupture [99]. In addition, ROS are generated from AuNPs, which can weaken the bacterial cell wall further by inducing oxidative damage. Combined with the photothermal effect, these processes lead to the eventual bacterial cell wall disruption and death [100]. Neurological infections pose a huge threat to human health and cause a lot of deaths. Once the bacteria cross the blood brain barrier (BBB), it can spread in the central nervous system, leading to severe immunological reactions and inflammation. A study by Rizvi et al. [101] discusses how AuNPs can be used to treat bacterial infections in the brain. The characteristics of AuNPs, such as biocompatibility, stability, size tunability, and surface affinity to specific functional groups, allow them to cross the blood–brain barrier to effectively treat neurological infections. In addition, AuNPs have a unique ability to kill pathogens by inhibiting multiple aspects of their physiology, but conjugating them with antibiotics on their surface increases their effectiveness. AuNPs can disrupt bacterial membranes by forming pores in the cell wall, leading to permeability. In addition, they can inhibit ATPase activity in bacteria, disrupting cellular energy metabolism and leading to bacterial death. AuNPs can also disrupt the bacterial transcription mechanism, inhibiting the synthesis of essential proteins, and can increase bacterial chemotaxis. Another property that makes AuNPs suitable for treating neurological infections is their ability to cross the BBB, as small-sized AuNPs with a negative charge have better penetration and are more likely to cross the BBB.

### 9.6. Cardiovascular Diseases Treatment

Cardiovascular diseases have always posed a huge threat to human health, and the existing drugs have severe side effects, as they require a lifetime of medication. Even though such drugs may delay symptoms, they fail to control the disease progression. Other solutions, such as cardiac transplantation, require surgery and are extremely invasive, with a high failure probability. Due to their low toxicity and ease of synthesis, AuNPs have been investigated for treating heart diseases such as atherosclerosis, heart failure, and endocarditis. A review by Zhang et al. [102] discussed the conjugation of AuNPs with clinical drugs for the treatment of heart diseases. Studies have shown that the accumulation of AuNPs injected intravenously in mice in the heart is size-dependent. AuNPs ranging from 10 to 250 nm have been found to accumulate in a healthy heart. Meanwhile, larger ones were detected mostly in the blood, liver, and spleen only. Anyways, studies have shown that larger AuNPs (30 nm–2 μm) are entrapped in diseased hearts, especially in injured ventricles or small heart capillaries. The downside of this is that larger AuNPs may lead to serious off-target effects, such as accumulating in the spleen. Conjugating AuNPs with clinical drugs for cardiovascular treatments has been promising in enhancing drug delivery. Spivak et al. [103] have shown positive results by treating DOX-induced heart failure rats with 30 nm AuNPs conjugated with Simdax, which was more effective than the Simdax treatment alone. Beta-blockers such as Metoprolol are good candidates for conjugation with AuNPs, as they accumulate more easily and quickly in the heart muscle and target abundantly expressed β1 receptors in the cells [104].

## 10. Limitations of the Usage of AuNPs in Drug Delivery

Even though AuNPs are considered generally safe and biocompatible as promising drug delivery agents due to their unique physicochemical properties, factors such as the NPs’ size, drug dosage, electrical charge, route of administration, and purity of the gold formation can play a role in determining the cytotoxicity. For example, the presence of free surfactants and metal ions other than gold particles in the solution can cause toxicity in biological environments. Regardless of the target site in the body, the liver and spleen are the main targets after exposure to AuNPs, as they are part of the immune system. These organs are responsible for the uptake, metabolism, and excretion of these NPs, which makes them a paramount topic for cytotoxicity analysis. In addition, the smaller the nanoparticle, the more widely it will be distributed in the body, so smaller AuNPs tend to cause more cytotoxicity for their size [105]. Having AuNPs accumulated in undesired places in the body due to low precision and inefficient targeting methods can defeat their purpose or cause hazardous side effects. Therefore, it is crucial to find a balance between the AuNPs’ size and payload to minimize off-target effects and stay within safety limits.

One of the main problems that medication targeting the central nervous system (CNS) faces is the inability to cross the BBB, which not only prevents drugs from passing but also disposes of them. As a result, NPs that deliver drugs to the CNS must be designed to overcome this challenge. Provoking permeability in the BBB by disrupting the tight junctions between its cells using US is one way to create a transient path for the NPs to pass. In addition, surface modifications, such as a positive charge, on the NPs can facilitate their binding to the negatively charged plasma membranes of the BBB cells. Factors such as size, charge, surface modifications, and circulation time all influence the ability of NPs to cross selective body membranes [106]. As mentioned in this paper, the size and shape of AuNPs must be precisely determined, as any slight inaccuracy leads to changes in biodistribution, cellular uptake, and therapeutic efficiency. So, achieving uniformity in their size and shape is a prerequisite to the reproducibility and reliability of AuNPs as drug delivery agents. Additionally, encapsulation strategies, surface modifications, and release mechanisms must be tailored to every specific application and physiochemical properties of the encapsulated drug. Release strategies must be carefully carried out to minimize off-target effects and achieve optimal therapeutic efficiency. Another concern is the AuNPs’ biological stability in physiological environments. Conditions such as temperature, pH, and the presence of certain ions, solutes, and chemicals in the body must not influence the toxicity, efficiency, and purpose of AuNPs, as bonds of encapsulated or loaded drugs must not be disrupted or broken until intended. Despite the challenges and careful considerations needed in utilizing AuNPs for drug delivery, their remarkable properties offer great opportunities that allow scientists to revolutionize them and understand their biological role in medical trials.

## 11. Conclusions

The remarkable versatility of AuNPs renders them innovative agents in several fields, most notably in drug delivery and therapeutics. This comprehensive review highlights the great impact of AuNPs on drug delivery systems, focusing on their unique properties that make them effective in this domain. Various aspects of AuNPs, such as their shape, functionalization, drug-loading bonds, targeting mechanisms, therapeutic effects, and cellular uptake methods, have been discussed in this work. AuNPs have been extensively studied for their potential in cancer treatment, as various in vitro and in vivo studies highlighted in the review. Given their broad therapeutic potential, AuNPs have a significant role in advancing drug delivery and improving treatments for a range of diseases, including cancer, diabetes, skin diseases, cardiovascular issues, and ocular diseases.

## Figures and Tables

**Figure 1 pharmaceutics-16-01332-f001:**
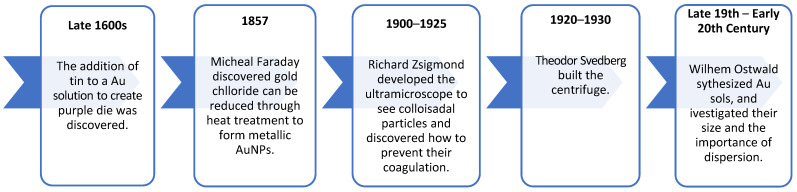
History of the development of AuNPs [16].

**Figure 2 pharmaceutics-16-01332-f002:**
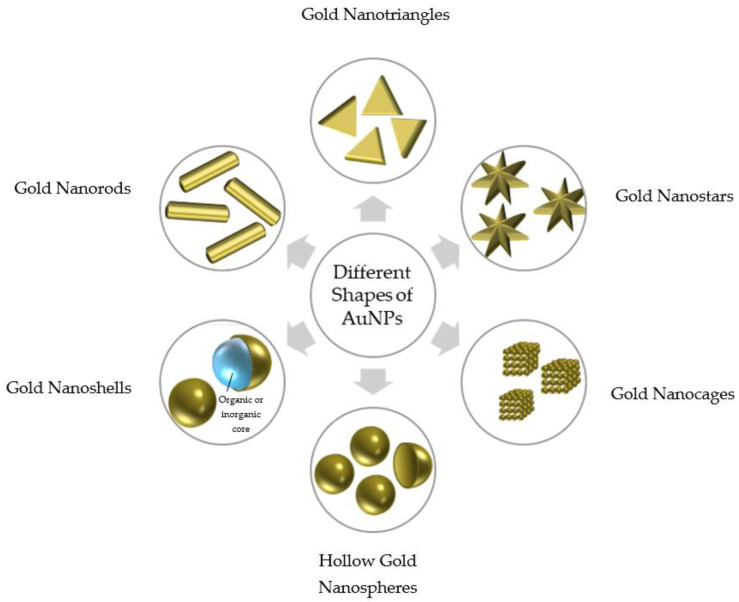
Different shapes of AuNPs.

**Figure 3 pharmaceutics-16-01332-f003:**
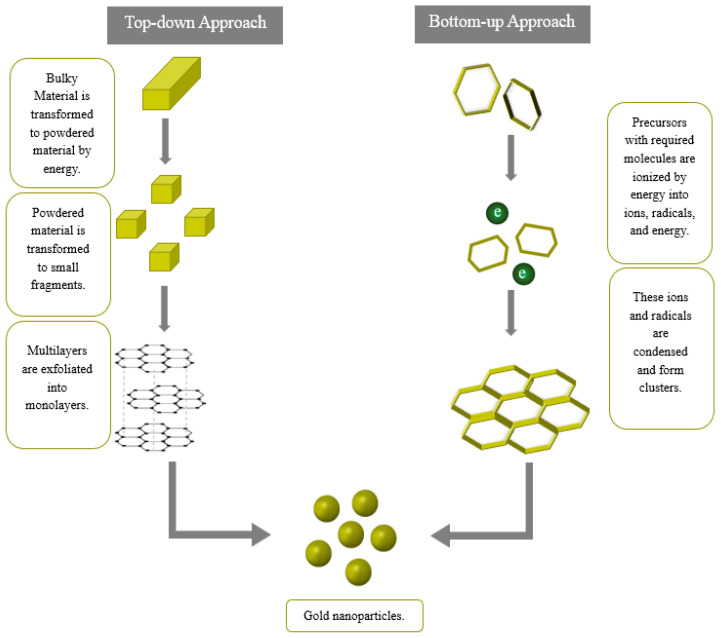
Top-down vs. bottom-up approach in the synthesis of AuNPs, adapted with changes from [44].

**Figure 4 pharmaceutics-16-01332-f004:**
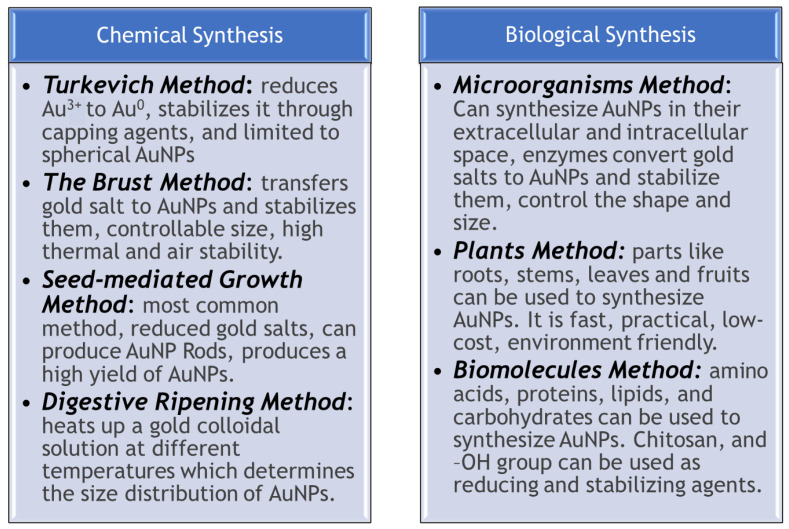
Summary of the chemical and biological AuNPs synthesis methods.

**Figure 5 pharmaceutics-16-01332-f005:**
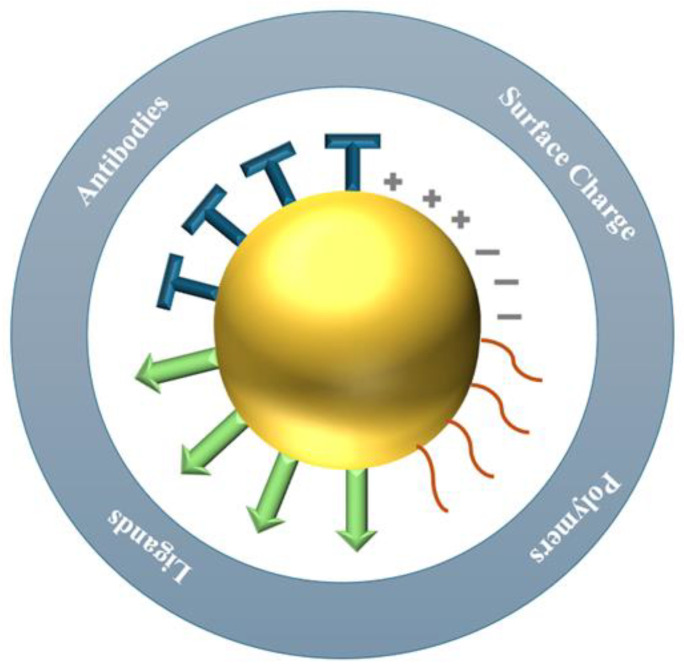
AuNPs’ surface modification.

**Figure 6 pharmaceutics-16-01332-f006:**
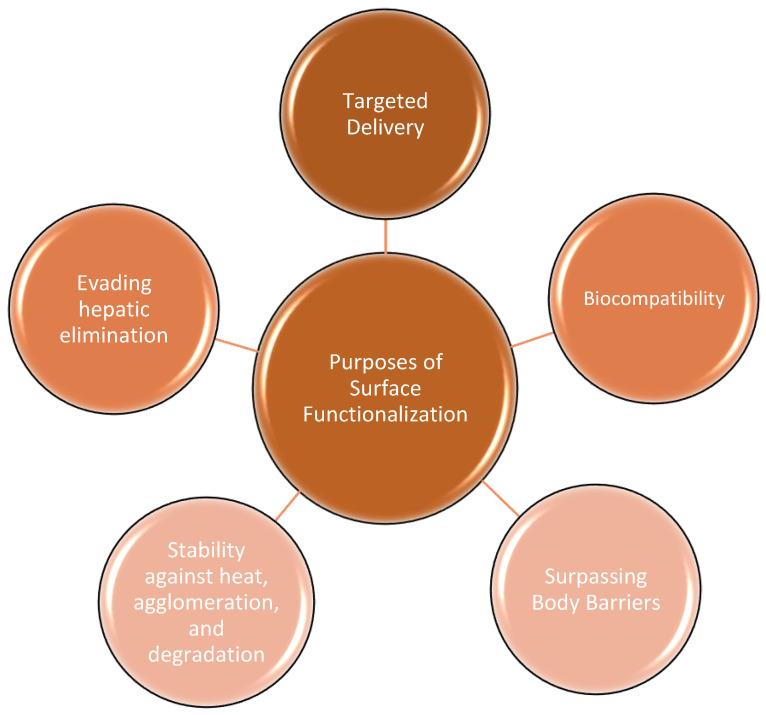
Benefits of AuNPs’ surface functionalization.

**Figure 7 pharmaceutics-16-01332-f007:**
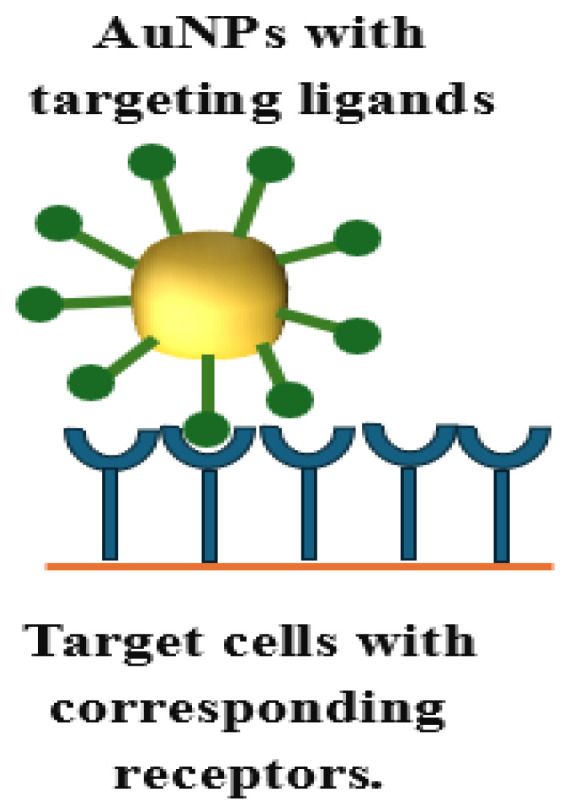
How AuNPs with ligands attach to receptors on target cells in the body.

**Figure 8 pharmaceutics-16-01332-f008:**
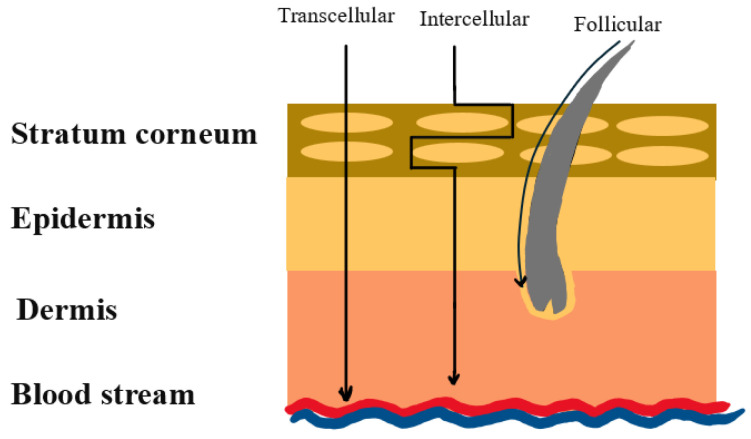
Different routes taken by AuNPs to penetrate the skin.

**Table 2 pharmaceutics-16-01332-t002:** Advantages and disadvantages of chemical and biological synthesis methods.

	Method	Advantages	Disadvantages
Chemical Synthesis	Turkevich Method	- Uncomplicated and reproducible	- Particles above 30 nm are less spherical and have a broader size distribution. - Low yield.
	The Burst Method	- AuNPs produced have a controllable size, less dispersity, and are thermal and air-stable.	- Their biological applications are limited.
	Seed-Mediated Growth	- Reliable method to produce gold nanorods.	- Factors like the concentration of HauCl4, temperature, and number of seeds must be highly controlled as they affect the size and aspect ratios.
	Digestive Ripening Growth	- Easy method to produce monodispersed NPs. - Produces a high yield.	- Controlling the size of NPs is very difficult.
Biological Synthesis	Microorganisms	- Some bacterial species are not affected by heavy metals. - Extracellular synthesis produces pure AuNPs.	- Slow and tedious process that can take anywhere from hours to days. - It is complicated to prepare biomass from fungi.
	Plants	- The process is facile, uncomplicated, fast, and economical. - AuNPs’ size and shape can be controlled by the reaction parameters.	- It is challenging to identify reactive components in plant biomass. - Algae takes a lot of time to grow.
	Biomolecules	- Contains a lot of functional groups that can aid in AuNPs synthesis.	- Each biomolecule’s reducing ability must be determined before using it in the synthesis reaction.

**Table 3 pharmaceutics-16-01332-t003:** Factors that influence the efficacy of passive tumor targeting by AuNPs.

Factors	Criteria
Size	NPs should be small enough to penetrate tissues effectively. They should also be big enough to be retained better in the tumor site and avoid elimination.
Surface Charge	Slightly negative or neutral surface charge is preferred to prevent aggregation and immediate elimination by the immune system. Having a positive charge when reaching the tumor will enhance drug uptake into the cell.
Shape	Shape plays a vital role in specificity. For example, rod-shaped NPs have higher efficiency in cell targeting due to their improved interaction with tissues. NPs’ shape affects cellular uptake mechanisms, efficiency, and surface adhesion. Also, shape affects their circulation in the blood and their interaction with immune cells, promoting a response from them.
Elasticity	Elasticity influences NPs’ circulation time and accumulation. Soft ones tend to be more flexible and accumulate more effectively in tumor sites than hard NPs.
Avoiding Opsonization	Using zwitterionic polymers and PEGs can prevent the rapid clearance of NPs by the immune system to prolong their circulation time.
Avoiding Reticuloendothelial System	Kupffer cells in the liver comprise the RES system, which is crucial for controlling the circulation time of NPs. Ligand chemistry, surface charge, and size must be optimized to avoid recognition and elimination by these immune cells.

**Table 4 pharmaceutics-16-01332-t004:** Advantages and disadvantages of intrinsic and extrinsic triggering methods of AuNPs.

Nature of Triggering Mechanisms	Triggering Methods	Advantages	Disadvantages
Extrinsic Methods	Light Triggering [75]	By adjusting laser intensity and duration of irradiation, the PTT can target AuNPs accumulated in cancer sites and minimize damage to healthy cells.	Necrosis, or cell death due to acute injury, may result from PPT, which induces an inflammation response in the body, causing discomfort to the patients. In addition, some cancer cells can develop immunity to apoptosis, or programmed cell death caused by the heat generated by AuNPs, reducing treatment efficiency.
	US Triggering [76]	The use of US to trigger drug release is not invasive, which eliminates painful surgeries or interventions. Also, US waves can be carefully controlled and focused on tumor sites. US waves can penetrate deeply into tissues.	Very low frequencies (<250 kHz) are very difficult to focus on small volumes, which will cause healthy tissue to be subjected to US and probably killed.
Intrinsic Methods	pH Triggering	pH-sensitive drug bonds on AuNPs can exploit the acidic nature of the tumor microenvironment. This allows for controlled release and minimal side effects.	Non-specific pH changes in the body might trigger the release of the drugs in off-target sites, minimalizing the specificity of AuNPs.
	Enzyme Activity Triggering	Enzyme-specific release is advantageous, as AuNPs can be designed to respond to enzymes unique to specific diseases, resulting in particular drug release.	Enzyme activity might differ for various diseases and individuals, which may involve complex NP development. In addition, target enzymes might exist in other areas of the body, triggering off-target drug release.
	Redox Reaction Triggering	Redox-responsive release can utilize redox reactions that occur in the specified target disease in the body.	The body’s redox environment might differ between cell types and physiological conditions. This variability might cause nonspecific drug release, causing off-target effects.

**Table 5 pharmaceutics-16-01332-t005:** AuNP in in vivo therapeutic applications in cancer.

AuNP Type	Shape and Size	Cancer Site	Drug	Loading Mechanism	Targeting Mechanism	Reference
FA–CurAu-PVP NPs	Size: 250 nm (average) Shape: Gold Nanospheres	Breast Cancer	Curcumin	The drug is loaded onto the surface of AuNPs	Folate receptor-mediated active targeting and pH-sensitive release	[78]
Anti-EGFR-HAuNS	Size: 30 nm Shape: hollow gold nanoshells	Any cancer cell with epidermal growth factor receptor (EGFR)	PTT/No drug	-	Active targeting through the functionalization of monoclonal antibody that targets EGFR.	[79]
DOX-NN-AuNPs and DOX-S-AuNPs	Size: 30 nm Shape: Gold Nanospheres	Oral cancer	Doxorubicin	thyl-thioglyconate and hydrazine compounds were used to chemically link DOX to AuNPs resulting in a pH-sensitive linker (S-).	Custom RGD peptide was coating the AuNPs to enhance active targeting of oral cancer cells.	[80]
DOX@AuNPs	Size: 4.70 ± 0.89 nm Shape: Gold Nanospheres	Cancer in general	Doxorubicin	DOX was loaded on PEG-AuNPs because of the π–π with 4-mercaptobenzoic acid (MBA) incorporated on the surface.	Passive Targeting	[81]
PEGylated AuNP-Pc4	Size: 3–7 nm Shape: Gold Nanospheres	Cancer in general	Silicon phthalocyanine 4 (Pc 4)	Pc 4 is attached to the AuNPs’ surface through N-Au bonding.	Passive Targeting	[82]
MGNPs-DOX	Size: 22 nm Shape: Gold Nanospheres	Carcinoma	Doxorubicin	The drug was conjugated on the surface of the AuNPs through electrostatic interaction between the DOX anime group and the surface.	Magnetic targeted delivery through an external magnetic field.	[63]
cRGD-uPIC-AuNP	Size: 20 nm Shape: Gold Nanospheres	Cervical Cancer	siRNA	cRGD peptide allows for active targeting.	The binding of the siRNA-cRGD-PEG-PLL-LA complex onto the AuNPs is through universal polyion complex (uPIC)	[83]
Apt-Au@MSL	The AuNPs’ size: 10 nm Shape: Gold Nanospheres	Cancer cells in general	Morin	Morin is encapsulated in the liposomes.	Aptamers on AuNPs, which are incorporated on the surface of liposomes, allow for active targeting.	[84]
Doxorubicin-Oligomer-AuNP (DOA)	Size: 18–28 nm Shape: Gold nanospheres	Colon Cancer	Doxorubicin	Dox was loaded on oligonucleotides, which are the AuNPs’ capping agents.	Passive Targeting	[85]
Disulfide cross-linked short polyethyleni-mines GNDs	Aspect ratio: 4.1 Shape: Gold nanorods	Brain and breast cancer	small hairpin (sh)RNA	shRNA is connected to the GND through GSH-triggered disulfide bond.	RGD ligand actively targeted the αvβɜ receptors on cancer cells. The NPs also utilized passive targeting.	[86]

**Table 6 pharmaceutics-16-01332-t006:** AuNP in in vitro therapeutic applications in cancer.

AuNP Type	Shape and Size	Cancer Site	Drug	Loading Mechanism	Targeting Mechanism	Reference
AS1411-AuNPs	Size: 18.3–21.16 nm Shape: Gold nanospheres	Cervical cancer	acridine orange derivative (C8) or Imiquimod (IQ)	Drug is associated via supramolecular assembly	Aptamer AS1411 covalently bonded to AuNPs is used for active targeting.EPR effect was utilized as well.	[77]
DOX-Fu AuNPs	Size: 73–96 nm Shape: Gold nanospheres	Breast cancer	Doxorubicin	The noncovalent hydrogen-bonding loading of the drug onto AuNPs	Active targeting through molecule binding and pH-sensitive release.	[87]
GNPs-L-Aspartate nanostructure	Size: - Shape: Gold Nanospheres	Liver Cancer	doxorubicin, cisplatin, and capecitabine	The drugs are non-covalently conjugated onto the aspartic acid assemblies on the AuNPs	Targeting ligands on AuNPs actively targeted overexpressed receptors on hepatocellular carcinoma cells.	[88]
Au@p12 + CRGDK	Size: 2 nm	Breast Cancer	Therapeutic PMI (P12) peptide	The therapeutic P1q2 peptide was conjugated on the surface of the AuNPs.	Target peptide CRGDK allows for selective binding to neuropilin-1(Nrp-1) and receptor-mediated endocytosis.	[89]
PEG-AuNPs	Size: 6.7 ± 0.5 nm Shape: Gold Nanospheres	Pancreatic Cancer	Doxorubicin	DOX is bonded through an amide bond (enzyme-sensitive) to the PEG layer.	Active targeting is through anti-Kv11.1 polyclonal antibody [pAb]. Release is through enzymatic cleavage of the amide bond.	[90]
Gold Nanoshells	Shape: Gold nanoshells Size: 140 nm	Liver Cancer	PTT/No drug	-	Active targeting through functionalization with 12-amino acid sequence peptides. NIR caused PT ablation.	[91]
PEG-Tf-NP	Size: 50.5 nm Shape: Gold Nanospheres	Breast Cancer	PTT/No drug	-	AuNPs are conjugated with transferrin molecules for active targeting. NIR caused PT ablation.	[92]
DOX-Hyd@AuNP	Size: 30 nm Shape: Gold Nanospheres	Addresses multidrug resistance in breast cancer cells.	Doxorubicin	DOX is conjugated to the AuNP through a pH-sensitive hydrazone bond.	Release is triggered by acidic (pH =5) environment	[93]
Anti-EGFR-HAuNS	Size: 30 nm Shape: hollow gold nanoshells	Any cancer cell with epidermal growth factor receptor (EGFR)	PTT/No drug	-	Active targeting through the functionalization of a monoclonal antibody that targets EGFR.	[79]
GNP-20-P1, GNP-20-P2, GNP-40-P1, and GNP-40-P2	Size: 20–40 nm	Prostate and Colon Cancer	Kahalalide F peptide (P1 and P2 analogs)	free thiol containing Cys, which facilitates the attachment to the AuNPs, is added to the peptide sequence.	The conjugation of P1 and P2 on the AuNPs induces a positive charge, which attracts them to negatively charged cell membranes and enhances absorption through receptor-mediated endocytosis.	[94]

## Data Availability

Not applicable.

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
