# Peer review of "Review of Gold Nanoparticles: Synthesis, Properties, Shapes, Cellular Uptake, Targeting, Release Mechanisms and Applications in Drug Delivery and Therapy"

_pharmaceutics, 2024, doi:10.3390/pharmaceutics16101332_

Round 1
Reviewer 1 Report
Comments and Suggestions for Authors
The review paper provides an overview of the therapeutic applications of gold nanoparticles. It covers a comprehensive range of topics, from the history of gold nanoparticles to synthesis and applications in disease treatment. I think that it would be helpful for readers who are not familiar with drug delivery and gold nanoparticles. However, since the review paper covers a wide range of things, it may seem a bit unfocused and include some sections that appear unnecessary. I would like to make a few suggestions regarding this paper.
1. Introduction covers a broad range of topics related to drug delivery. The characteristics of nanoparticles presented in Table 1, as well as the various types of nanoparticles discussed in Table 2, do not seem to be particularly important in the context of this review paper.
2. I am not sure if the detailed description of the history of gold nanoparticles is essential to the overall flow of this review paper.
3. The properties of gold nanoparticles are relatively important and I suggest that you put more detailed explanation. It would also be helpful to mention the characteristics of gold nanoparticles in comparison to other nanoparticles.
4. The shapes of gold nanoparticles can affect characteristics of gold nanoparticles. Instead of merely describing the shapes in text, it would be more effective to present them in a figure for better understanding.
5. I am not sure if it is necessary to separate the AuNPs targeting mechanisms into sections such as active targeting, passive targeting, and magnetic targeting.
6. The text transitions directly from section 7.3 on passive targeting to section 8.3 on photochemical release. It seems that some intermediate sections may be missing. Please check to ensure that no content has been omitted.
7. The table referenced as Table 1 on page 21 should be renumbered to Table 5, and the table referred to as Table 2 on page 23 should be corrected to Table 6.
8. The numbering of Figure 2 on page 27 seems to be incorrect and should be revised. Among the various topics in section 9, it seems unnatural that only the '9.2 skin disease treatment' includes a figure. Moreover, the content of the figure is just a general illustration of skin tissue and does not seem to be important.
Comments on the Quality of English LanguageThe use of English in the paper does not seem to have significant issues.
Author Response
Please find the attached file with the rebuttal to both reviewers. Please note that I was not able to paste the figures into the designated space.

Reviewer 2 Report
Comments and Suggestions for Authors
The authors review various aspects of gold nanoparticles, starting from general properties and synthesis techniques up to applications in diagnostics and therapy, including their limitations.
Gold is an important material for nanoparticle applications in the life sciences due to its many favorable properties, which the authors nicely discuss. Therefore, I recommend the publication of the article in Pharmaceutics after the authors have considered my issues below.
Major issues:
- The list in lines 91 to 140 on various nanoparticles does not fit the manuscript's title, indicating that the focus is _gold_ nanoparticles. Furthermore, most points are based on a single article, Ref. [15]. The list should be removed from the manuscript.
- Table 2 contains information on gold nanoparticles only in the 9th of the in total nine rows. Furthermore, most of the information is based on Ref. [15]. The table should be removed from the manuscript.
- In several places, the authors mention "small" and "large", or even do not provide the size of the nanoparticle. Because gold nanoparticles have a wide range of sizes that interact in a qualitatively different way with lipid-bilayer membranes, the authors should provide size ranges instead of qualitative terms. This comment applies to, for example, line 190, lines 200ff, line 244, line 260, line 263, lines 268ff., and line 370. Because of the importance of nanoparticle size, I suggest the authors carefully check the entire text for missing data on the sizes.
- Lines 190/191: The statement that small gold nanoparticles are internalized via endocytosis is incorrect. Single particles with sizes of the order of the thickness of the lipid bilayer (~4 nm) can translocate through the bilayer, whereas larger particles (> 20 nanometers) get wrapped/endocytosed. See, e.g., https://doi.org/10.1116/1.5022145.
- Lines 218/219: Particle shape is mentioned in the title and, thus, a main theme of this manuscript--and gold nanoparticles interact in two fundamentally different manners with lipid bilayers depending on the particle size. Therefore, I suggest that the authors discuss these two mechanisms, e.g., based on these two articles "Robert Vácha, Francisco J. Martinez-Veracoechea, and Daan Frenkel. "Receptor-mediated endocytosis of nanoparticles of various shapes." Nano letters 11.12 (2011): 5391-5395." and "Sabyasachi Dasgupta, Thorsten Auth, and Gerhard Gompper. "Shape and orientation matter for the cellular uptake of nonspherical particles." Nano letters 14.2 (2014): 687-693.".
- Figure 1: Do gold atoms typically form hexagonal rings? If this is the case, I suggest that the authors mention this also in the main text and cite an appropriate reference. If this is not the case, I suggest that the authors revise the figure.
- Lines 900/901: It may also be a buildup of membrane tension that disrupts the cell wall, see, e.g., https://doi.org/10.1016/j.ijvsm.2017.02.003, https://doi.org/10.1021/acs.nanolett.9b04788, and https://doi.org/10.1002/smll.201200528.
Minor issues:
- Line 65: consider replacing the word "things" with "particles" or "structures"
- Table 1: Is traditional DDS really applied to the entire body?
- Lines 162-176: The entire paragraph mentions many different facts but contains only a single reference. This appears not appropriate.
- Line 511: functionalize it -> functionalize them
- Line 634: Is shape relevant for targeting or only for uptake/translocation?
- Line 706: The sentence "z due to enzymic activity" is incomplete.
- Line 818: Reference needed. Can the authors add a sentence to describe why shape is crucial for dermal and transdermal drug delivery applications and discuss a few shapes?
- Line 948: What does "US" stand for?
- Lines 978-1039: These lines should be deleted.
none
Author Response
Please find the rebuttal for the reviewer's' comments. Please note that I was not able to paste the answers because they have figures.

Round 2
Reviewer 1 Report
Comments and Suggestions for Authors
You provided a clear response to the revisions I suggested.
I have no further suggestions to offer.